The modified Thomas test is not a valid measure of hip extension unless pelvic tilt is controlled

Vigotsky Andrew D. avigotsky@gmail.com 1
Lehman Gregory J. 2
Beardsley Chris 3
Contreras Bret 4
Chung Bryan 5
Feser Erin H. 1
1 Kinesiology Program, Arizona State University , Phoenix , AZ , United States
2 Private Practice , Toronto , Ontario , Canada
3 Strength and Conditioning Research Limited , London , United Kingdom
4 School of Sport and Recreation, Auckland University of Technology , Auckland , New Zealand
5 Department of Plastic Surgery, Island Health Authority , Victoria , British Columbia , Canada
Keogh Justin
Electronic publication date: 2016 Aug 11
Publication date: 2016
Volume: 4
Electronic Location ID: e2325
Received 2016 May 24; Accepted 2016 Jul 14
Copyright: ©2016 Vigotsky et al.
Copyright year: 2016
Copyright holder: Vigotsky et al.
License: This is an open access article distributed under the terms of the Creative Commons Attribution License, which permits unrestricted use, distribution, reproduction and adaptation in any medium and for any purpose provided that it is properly attributed. For attribution, the original author(s), title, publication source (PeerJ) and either DOI or URL of the article must be cited.
License URL: https://creativecommons.org/licenses/by/4.0/

Keywords: Orthopedic testing, Orthopaedic testing, Hip mobility

Funding: The authors received no funding for this work.

==============================
The modified Thomas test was developed to assess the presence of hip flexion contracture and to measure hip extensibility. Despite its widespread use, to the authors’ knowledge, its criterion reference validity has not yet been investigated. The purpose of this study was to assess the criterion reference validity of the modified Thomas test for measuring peak hip extension angle and hip extension deficits, as defined by the hip not being able to extend to 0º, or neutral. Twenty-nine healthy college students (age = 22.00 ± 3.80 years; height = 1.71 ± 0.09 m; body mass = 70.00 ± 15.60 kg) were recruited for this study. Bland–Altman plots revealed poor validity for the modified Thomas test’s ability to measure hip extension, which could not be explained by differences in hip flexion ability alone. The modified Thomas test displayed a sensitivity of 31.82% (95% CI [13.86–54.87]) and a specificity of 57.14% (95% CI [18.41–90.10]) for testing hip extension deficits. It appears, however, that by controlling pelvic tilt, much of this variance can be accounted for (r = 0.98). When pelvic tilt is not controlled, the modified Thomas test displays poor criterion reference validity and, as per previous studies, poor reliability. However, when pelvic tilt is controlled, the modified Thomas test appears to be a valid test for evaluating peak hip extension angle.

Introduction

The Thomas test (TT), named after Dr. Hugh Owen Thomas, was created to rule out hip flexion contracture (Thomas, 1878), meaning that a positive TT is indicative of hip flexion contracture. Since then, it has been used ubiquitously to assess hip extensibility. The TT is a pass/fail test in which the patient lies supine upon an examination table with both legs straight out in front of them on the table top. While supine, the patient flexes the hip of one leg and holds the knee of the same leg maximally flexed at the chest. The pelvis is maintained in neutral throughout. The contralateral leg is allowed to remain relaxed and flat against the tabletop. A positive TT, which is taken as indicative of hip flexion contracture, is where there is noticeable hip flexion of the contralateral leg, as indicated by a gap between this leg and the table top. For the purposes of this study, the aforementioned hip flexion contracture will be referred to as a hip extension deficit, as more than just contracture can inhibit hip extension. The modified TT (MTT) is performed in a similar fashion to the original Thomas test, but is carried out at the edge of the tabletop. Thus, the contralateral leg is allowed to hang down over the edge of the table, which permits the measurement of a peak hip extension angle in all individuals and not just those with a hip extension deficit.

There are numerous potential confounders with both the TT and MTT that may yield them invalid for their intended purpose. Most importantly, they do not consider lumbopelvic movement, hip flexion ability, waist size, or thigh circumference. Lumbopelvic movement may influence the outcome of the MTT in two ways, in that anterior pelvic tilt can mimic hip extension, thus rendering a false negative or inflated peak hip extension angle, or vice-versa with posterior pelvic tilt. Presumably, lumbopelvic movement is at least partially due to hip flexion ability of the hip contralateral to the one being tested, or how much hip flexion range of motion (ROM) one possesses before his or her pelvis is forced to rotate. A restriction in hip flexion ability will force a person into a posterior pelvic tilt when trying to bring his or her knee to his or her chest; however, a person with substantial hip flexion ability will be able to perform simultaneous anterior pelvic tilt, thus potentially rendering a false negative or inflated peak hip extension angle. Waist size and thigh circumference are separate from, but have similar effects as, hip flexion ability. A person with large thigh and waist circumferences may not be able to exhaust his or her hip flexion ability before his or her thigh and waist make contact, which will allow for that person to utilize anterior pelvic tilt during testing.

Hip extension is considered to be important for the performance of various athletic activities. A lack of hip extension has been theorized to lead to an overstriding gait and increased impact forces during running (Derrick, Hamill & Caldwell, 1998; Franz et al., 2009), which may increase the risk of tibial stress fracture (Edwards et al., 2009). Further, a lack of hip extension may be associated with tightness in the hip flexor muscles. A postural hypothesis related to hamstring strains is that tight hip flexors lead to an anterior pelvic tilt, which may predispose sprint athletes to hamstring strains (Gabbe, Bennell & Finch, 2006). Lastly, for individuals with low back pain that is sensitive to spinal extension, tight hip flexors may lead these individuals to perform spinal movements that bias increased spinal extension, as the individual lacks movement options due to their hip extension limitations.

The reliability of both the TT and MTT has been studied with mostly positive outcomes outcomes (Aalto et al., 2005; Cejudo et al., 2015; Clapis, Davis & Davis, 2007; Gabbe et al., 2004; Harvey, 1998; Heino, Godges & Carter, 1990; Lai et al., 2012; Parikh & Arora, 2015; Peeler & Anderson, 2007a; Peeler & Leiter, 2013; Peeler & Anderson, 2007b; Petersen et al., 2015; Pua et al., 2008; Roach et al., 2013). However, to the authors’ knowledge, only the TT has been validated, which was shown to have convergent validity with maximum hip extension during stance phase of gait, hip flexor index, and maximum psoas length in normal controls, but not in patients with cerebral palsy (Lee et al., 2011). Therefore, the purpose of this investigation was to determine the criterion reference validity of the MTT using more objective measures; namely, two-dimensional sagittal plane motion capture measured relative to the pelvis.

Methods

Participants

Healthy participants were recruited from a student population via flyers placed around campus and presented to Kinesiology and Exercise and Wellness classes. Before each participant was scheduled for testing, investigators asked the participant about his or her current injury status. Participants were excluded if they had current symptoms of back or lower extremity musculoskeletal or neuromuscular injury or pain; however, participants were not excluded if they previously had a back or lower extremity musculoskeletal or neuromuscular injury but were currently symptom-free, no matter how recently symptoms may have been experienced. Participants were scheduled to come into the laboratory for one visit. Upon arrival, participants were provided a verbal explanation of the study, and read and signed an Informed Consent and Physical Activity Readiness Questionnaire (PAR-Q) before beginning. Any participant that answered “Yes” to any of the questions on the PAR-Q was excluded. The study was approved by the Institutional Review Board at Arizona State University (IRB ID: STUDY00001660).

Preparation and measurement

After completing an Informed Consent and PAR-Q, participants’ age, height, and body mass were measured (Table 1). Thereafter, a ten-minute standardized warm up procedure followed. This warm up consisted of five minutes on an Airdyne bike, two sets of 20 bodyweight squats, two sets of 10 leg swings in both the frontal and sagittal planes, and two sets of 10 bodyweight lunges (Vigotsky et al., 2015).

Table 1 Descriptive statistics of participants.

Sex	n	Age (years)	Height (m)	Body mass (kg)	
Male	11	22.18 ± 4.14	1.79 ± 0.06	85.00 ± 10.00	
Female	18	21.80 ± 3.68	1.65 ± 0.06	60.71 ± 10.02	
Total	29	22.00 ± 3.80	1.71 ± 0.09	70.00 ± 15.60	
Notes.

Age, height, and body mass are presented as mean ± SD.

Once the ten-minute warm-up was completed, reflective markers were adhered to participants’ skin or tight fitting garments on the iliac crest, in line with the PSIS and ASIS and spaced 10 cm apart, the lateral femoral epicondyle, and the greater trochanter. These methods differ slightly from those presented by Kuo, Tully & Galea (2008), as the PSIS and ASIS markers were placed closer to the midaxillary line so as not to be blocked by the table or thigh during hip flexion (Vigotsky et al., 2015) (Fig. 1). True hip flexion and extension values were calculated by subtracting the four-point angles these markers create from 90°, as described by Sprigle et al. (2002) and Sprigle et al. (2003). Pelvic tilt was calculated as the angle between the intercristal line (created from the ASIS to PSIS) and horizontal plane, offset by 90°. Two-dimensional sagittal plane motion capture was obtained using an infrared camera set to 30 Hz (Basler Scout scA640-120; Basler Vision Technologies, USA) and motion analysis software (MaxTRAQ 2D; Innovision Systems Inc., USA).

Figure 1 Hip extension calculations.

The illustrated participant would have a hip extension angle of 8.1°(98.1º–90°). Illustration credit: Ji Sung Kim. From Vigotsky et al. (2015).

Procedures

The MTT was performed by having the participant hold his or her non-testing knee (left) to his or her chest, while letting the thigh and leg of the testing hip (right) hang freely (Harvey, 1998). However, the methods utilized for measuring true hip extension (motion capture) differ substantially from those previously described (Harvey 1998), in that the hip angle was measured relative to the pelvis rather than the plinth. This prevented lumbar hyperextension, decreased hip flexion ability, or large waist and thigh circumferences from confounding the results of the true hip extension test. Hip extension angles could then be compared relative to the pelvis (true hip extension) versus hip extension as it is typically measured with the MTT (thigh relative to the plinth). Each participant completed the MTT three times. Between each trial, the participant stood up from, and sat back down on, the plinth, as to “reset” his or her position. The average of each participant’s three trials was then used for analyses.

Statistical analyses

Bland–Altman plots, with 95% limits of agreement and 95% confidence intervals for those limits of agreement (Bland & Altman, 1986; Carkeet, 2015; Sedgwick, 2013), were created to determine the magnitude and variability of the differences between true hip extension and the MTT (that is, the angle of the thigh relative to horizontal), in addition to correlations. Pearson correlation coefficients were used to explore the possible source of discrepancy between true hip extension angle and the MTT, utilizing the difference between true hip extension and the result of the MTT and the following: hip flexion ROM before posterior pelvic tilt or thigh-waist contact; the sum of waist and thigh circumferences; and pelvic tilt during the MTT.

Figure 2 Bland–Altman plot of true hip extension and the modified Thomas test.

A mean difference of 0.7o, with 95% limits of agreements of −18.3o–19.7o, was found between the modified Thomas test and true hip extension. The black, solid line is indicative of the mean difference, whereas the black, dashed lines are indicative of the 95% limits of agreement. The blue, diagonal lines represent the 95% confidence intervals of the 95% limits of agreement. MTT, modified Thomas test; pink, female; blue, male.

The binary pass/fail outcome of a MTT is often determined by whether or not the thigh is above horizontal (Clapis, Davis & Davis, 2007; Ferber, Kendall & McElroy, 2010). In order to determine the validity of the MTT for determining the presence of hip extension deficits, the sensitivity, specificity, and their 95% confidence intervals were also determined. A test was said to be positive if, for the MTT, the thigh was above parallel (that is, if the knee was higher than the hip), or if, for the true hip extension test, a hip angle of ≥0°could not be obtained.

Results

Twenty-nine healthy participants were recruited for this study (Table 1). A Bland–Altman plot of MTT and true sagittal plane hip extension is shown in Fig. 2, and the raw data and differences between the MTT and true sagittal plane hip extension can be found in Table 2. The angle of the thigh relative to horizontal was moderately correlated with sagittal plane hip extension (r = 0.50). Correlations revealed that these differences could not be explained by hip flexion ROM alone (r = 0.11) or waist and thigh circumferences (r = − 0.12). In contrast, pelvic tilt was strongly associated with the difference between true hip extension and the MTT (r = 0.98) (Fig. 3). When assessing pass/fail for hip extension deficit, the MTT displayed a sensitivity of 31.82% (95% CI [13.86–54.87]) and a specificity of 57.14% (95% CI [18.41–90.10]).

Table 2 Raw values of, and differences between, true hip extension and the MTT.

	Sex	True (°)	MTT (°)	Δ (°)	
1	F	−1.1	19.4	20.5	
2	F	5.3	9.5	4.1	
3	M	−3.5	−2.9	0.6	
4	F	2.7	8.5	5.8	
5	M	4.0	5.0	1.0	
6	F	−5.3	−4.5	0.8	
7	F	3.6	8.9	5.2	
8	M	20.5	14.7	−5.8	
9	M	−4.5	−10.1	−5.6	
10	F	−0.6	2.3	2.9	
11	M	17.0	−15.4	−32.4	
12	M	−5.5	1.1	6.5	
13	F	−4.5	−5.7	−1.1	
14	M	−3.1	12.3	15.5	
15	F	16.3	11	−5.3	
16	F	10.8	1	−9.8	
17	F	2.7	−1.0	−3.7	
18	F	10.2	12.9	2.7	
19	F	8.9	7.1	−1.8	
20	F	12.1	6	−6.1	
21	F	9.2	15.7	6.5	
22	F	5.2	6.9	1.7	
23	F	−3.6	−10.2	−6.6	
24	M	12.1	2.1	−10.1	
25	M	−21.1	−19.2	2.0	
26	F	11.3	−6.4	−17.7	
27	M	−10.7	−10.9	−0.2	
28	M	0.3	3.5	3.2	
29	F	12.9	18.4	5.5	
x ¯		3.5 ± 9.2	2.8 ± 10.1	0.7 ± 9.7	
Notes.

Abbreviations True true hip extension

MTT modified Thomas test

Δ MTT − True

Figure 3 Difference between the modified Thomas test and true hip extension versus pelvic tilt during the modified Thomas test.

(−), posterior pelvic tilt; (+), anterior pelvic tilt; difference, modified Thomas test—true hip extension; pink, female; blue, male.

Discussion

Although the MTT is widely used in orthopedic and physiotherapy practice, its criterion reference validity has not previously been investigated. In this present study, the criterion reference validity of the MTT in testing hip extension was evaluated. It was found that, when compared to sagittal plane motion capture, the MTT was a relatively poor measure of hip extension (Fig. 2). However, pelvic tilt alone likely accounts for the variance between the MTT and true hip extension, suggesting that results recorded in the MTT are substantially affected by pelvic tilt. Additionally, when compared with sagittal plane motion capture, the MTT was also found to have poor specificity and sensitivity for determining hip extension deficits. None of these findings appear to be sex-dependent (Figs. 2 and 3).

The reported hip extension angles are not unlike those reported by Moreside & McGill (2011), who also evaluated hip extension using motion capture. The angles of the thigh relative to horizontal presented by Moreside & McGill (2011) appear to be different, though, as the authors used a pressure cuff under the lumbar spine to control for lumbopelvic movement and hip flexion differences. More specifically, the authors placed a blood pressure cuff, inflated to 60 mmHg, under participants’ lumbar spine, and if cuff pressure changed, it was indicative of lumbopelvic motion. Furthermore, the authors offset the MTT results by 10°, which assumes equal pelvic tilt is occurring for all participants. Our findings indicate that if pelvic tilt is corrected for, the discrepancies between the results of the MTT, true hip extension, and the MTT results reported by Moreside & McGill (2011) should be diminished.

Although the MTT has previously been assumed to be a test for hip extension ROM, this is not necessarily the case. ROM testing is typically performed either actively or passively; the former requiring the person in question to move the joint in question actively, with moments produced by his or her muscles, while the latter implies that an external force (such as a practitioner) generates a moment about the joint. In both active and passive ROM testing, typically, the ROM is taken to what is perceived as “end range.” However, as briefly noted by Zafereo et al. (2015) and Vigotsky et al. (2015), the MTT may not reflect true ROM endpoints; rather, it is posited that, because the only external force applied to the lower extremity is the weight of the limb itself, the external hip extension moment should be the same for all intraindividual tests. Should the hip extension moment be the same for each test, only a decrease in passive stiffness of the tissue being stressed (i.e., rectus femoris) would allow for an increase in the measured ROM; therefore, the MTT may be a measure of passive stiffness for one point in the individual’s ROM.

The findings of this present study are complementary to the reliability data reported by Kim & Ha (2015), who found that that the MTT is more reliable after correcting for lumbopelvic movement. Such a consideration has been previously suggested by other studies (Moreside & McGill, 2011), but until now, its importance has not been quantified. Moreover, the low sensitivity and specificity observed in this study have remarkable clinical implications, in that they suggest that practitioners who utilize the MTT to assess the presence of hip flexion contracture or a hip extension deficit, without controlling for pelvic tilt, are doing so with a high risk of both false positive and false negative findings. However, these data also suggest that the observed sensitivity and specificity can be drastically improved by controlling for pelvic tilt (Fig. 3). Future studies should investigate the effects on criterion reference validity of using different methods, such as palpation and inflatable cuffs, to control for pelvic tilt during the MTT.

The ASIS and PSIS references utilized in this study are just one method of measuring hip extension. Other methods exist to measure pelvic tilt, or hip extension, such as forming a (vertical) plane using the left and right ASIS and pubic symphysis (Kendall et al., 1993) or by creating a (horizontal) plane using the ischial spine and pubic symphysis (Sinnatamby, 2011). Such methods have been shown to produce different results from the ASIS-PSIS references utilized in this study (range = 0–23°; mean = 13 ± 5°) (Preece et al., 2008). However, such methods are not clinically applicable, and additionally, there is no consensus as to the exact definition and position of a “neutral hip.”

Conclusions

The data presented in this study suggest that the MTT is not a valid measure of hip extension unless lumbopelvic movement is controlled for. Specifically, the MTT displays poor sensitivity, specificity, and criterion reference validity relative to sagittal plane motion capture; however, much of this variance is due to pelvic tilt during the test. Due to the ubiquity of the MTT, the findings of this current study are highly relevant to the practice of musculoskeletal practitioners. It is of the utmost importance that, when utilizing the MTT, practitioners control for lumbopelvic movement in order to obtain a valid measure of peak hip extension angle or to identify the presence of hip flexion contracture.

Supplemental Information

Supplemental Information 1 Dataset for modified Thomas test validity paper

Click here for additional data file.

Additional Information and Declarations

Competing Interests

Author Contributions

Human Ethics

Data Availability

Chris Beardsley is a Director of Strength and Conditioning Research Limited, London, UK.

Andrew D. Vigotsky conceived and designed the experiments, performed the experiments, analyzed the data, wrote the paper, prepared figures and/or tables, reviewed drafts of the paper.

Gregory J. Lehman conceived and designed the experiments, reviewed drafts of the paper.

Chris Beardsley wrote the paper, reviewed drafts of the paper.

Bret Contreras contributed reagents/materials/analysis tools, reviewed drafts of the paper.

Bryan Chung analyzed the data, reviewed drafts of the paper.

Erin H. Feser performed the experiments, contributed reagents/materials/analysis tools, reviewed drafts of the paper.

The following information was supplied relating to ethical approvals (i.e., approving body and any reference numbers):

Arizona State University Institutional Review Board

IRB ID: STUDY00001660.

The following information was supplied regarding data availability:

The raw data has been supplied as a Supplemental Dataset.

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
