# Peer review of "The modified Thomas test is not a valid measure of hip extension unless pelvic tilt is controlled"

_PeerJ, doi:10.7717/peerj.2325_

## Round 0.1 · original submission · Minor Revisions

All three reviewers find much merit in the paper and have suggested some minor revisions. If the authors can adequately respond to these suggested revisions, this paper can be strongly considered for publication in PeerJ.

·

Basic reporting

Basic Reporting was all done well.
P2L23 & Abstract - Incorrect tense " was"
P2L40: There are more uses for the TT which may be important to mention.
Table 1: Caption requires "data presented as mean +/-SD"
Figure 2 - Discrete values for the mean and 95%CI may be helpful for the reader.
Figure 3 - Adding the r value to the plot may be helpful to the reader.
P11L288 - Incomplete reference
General - Ensure continuity of headings, abstract & methods capitalized, others lower case.

Experimental design

Experimental Design is clear and meaningful and is of a sound technical standard.

Validity of the findings

Data is robust and conclusions are appropriate. No Comments.

Comments for the author

This article is well written and adds value to the field. I commend them on their efforts.

·

Basic reporting

Introduction frames the purpose of the study well.

Experimental design

Methods section
Further explanation is needed in the methods section to clarify the exact procedure the patients underwent. i.e. exact set up/ testing procedure/ how many times the movement was performed/ how was pelvic tilt accounted for?

Validity of the findings

Results:
Report the mean difference between the MTT and True hip extension
Bland Altman plots should show mean difference and confidence limits
Run a one sample t-test to determine the mean difference between the two testing procedures is significant. This is assuming that the null hypothesis would result in mean difference of zero.
Discussion:
172-173: Remove the words ‘it is not’ and reword this sentence.
Conclusion:
Further development needed here. What do these findings mean for clinical practice.

Comments for the author

Overall this is a well conducted study which needs some minor revisions.

Line 57: ensure you clarify which hip you are referring to. i.e. in this instance you are referring to the contralateral hip which is being tested.
Line 92: specify how long they needed to be symptom free
Perhaps use sub-headings on line 115 ‘Procedure’ and before the paragraph starting on line 125 ‘Statistical Analysis’

·

Basic reporting

Introduction
Lines 62-63: “Waist size and thigh circumference are separate from, but similar to, hip flexion ability.” Please re-word this sentence. A physical dimension is not the same or similar to a physical ability.

Line 80: It appears there has been at least one paper that has assessed the validity of the TT using 3D motion analysis (J Bone Joint Surg Am. 2011 Jan 19;93(2):150-8. doi: 10.2106/JBJS.J.00252.) Please include this reference in your introduction accordingly and reword “However, to the authors’ knowledge, neither the TT nor MTT have been validated”. I would like to note that I have no connection to this paper in anyway and I am not trying self-promote.

Experimental design

Methods
Lines 86-87: “via flyers placed on campus” is odd wording, consider “via flyers placed AROUND campus”
Line 98: :change weight to mass and consider referring to your table of results for these data.
Please also present participant height in metres not centimetres in keeping with SI units.

Lines 99 and 103: Both “ten minute” and “10-minute” are used. Please choose one for consistency.

Line 111: Was 30 Hz used for data capture? If so, please just state that not also the camera capability so as to avoid confusion.

Validity of the findings

Results
Please make clear what dependant variable are being measured on the methods section as when I start reading the results and mention is made “table-femur angle”, for example, this has not been previously described.

Discussion
Lines 154-155: “its criterion reference validity has not previously been investigated.” As per comments above please revise this statement in light of the reference provided above.

Line 175: “moments produced from his or her muscles” consider “moments produced BY his or her muscles”

Line 187: “Such a consideration as been previously”, should this be “Such a consideration HAS been previously”

I am surprised that the poor sensitivity and specificity results have not been discussed. If they are not discussed then they should be presented. However, they make for a potentially interesting finding given their poor results for a test that is commonly used clinically. Therefore I suggest they should be presented and discussed.

Why have male vs female results not been discussed?

Comments for the author

The paper presented shows a clear and straightforward assessment of a commonly used clinical test. The paper is written clearly and succinctly and as such my comments are relatively minor.

---

## Round 0.2 · accepted · Accept

Thank you very much for your attention to detail in that the three reviewers are all happy with the amendments you've made to the original manuscript and have recommended that your manuscript be accepted for publication.

·

Basic reporting

I am happy with all the minor changes.

Experimental design

I am happy with all the minor changes.

Validity of the findings

I am happy with the minor changes

Comments for the author

Well done on your changes and the manuscript as a whole. Good job address all comments from reviewers.

·

Basic reporting

I am happy with the changes

Experimental design

I am happy with the changes

Validity of the findings

I am happy with the changes

Comments for the author

I am happy with the changes

·

Basic reporting

My previous comments have been adequately addressed.

Experimental design

My previous comments have been adequately addressed.

Validity of the findings

My previous comments have been adequately addressed.

Comments for the author

My previous comments have been adequately addressed.